# Prediction Value of Initial Serum Levels of SERPINA3 in Intracranial Pressure and Long-Term Neurological Outcomes in Traumatic Brain Injury

**DOI:** 10.3390/diagnostics14121245

**Published:** 2024-06-13

**Authors:** Haoyuan Tan, Jiamian Wang, Fengshi Li, Yidong Peng, Jin Lan, Yuanda Zhang, Dongxu Zhao, Yinghui Bao

**Affiliations:** 1Department of Neurosurgery, Ren Ji Hospital, Shanghai Jiao Tong University School of Medicine, Shanghai 200127, China; navneu@outlook.com (H.T.); jmwang0203@163.com (J.W.); lanjin@renji.com (J.L.); 2Neurologic Surgery Department, Huashan Hospital, Fudan University, Shanghai 200437, China; leafengshi@163.com; 3Brain Injury Center, Ren Ji Hospital, Shanghai Jiao Tong University School of Medicine, Shanghai Institute of Head Trauma, Shanghai 200127, China; pengyidong98@163.com; 4Minhang Hospital, Fudan University, Shanghai 200437, China; yd_zhang123@126.com

**Keywords:** SERPINA3, traumatic brain injury, rehabilitation, intensive care unit

## Abstract

Traumatic brain injury (TBI) is a severe neurological condition characterized by inflammation in the central nervous system. SERPINA3 has garnered attention as a potential biomarker for assessing this inflammation. Our study aimed to explore the predictive value of postoperative serum SERPINA3 levels in identifying the risk of cerebral edema and its prognostic implications in TBI. This study is a prospective observational study, including 37 patients with TBI who finally met our criteria. The Glasgow Outcome Scale (GOS), Levels of Cognitive Functioning (LCF), Disability Rating Scale (DRS), and Early Rehabilitation Barthel Index (ERBI) scores at six months after trauma were defined as the main study endpoint. We further calculated the ventricle-to-intracranial-volume ratio (VBR) at 6 months from CT scans. The study included patients with Glasgow Coma Scale (GCS) scores ranging from 3 to 8, who were subsequently categorized into two groups: the critical TBI group (GCS 3–5 points) and the severe TBI group (GCS 6–8 points). Within the critical TBI group, SERPINA3 levels were notably lower. However, among patients with elevated SERPINA3 levels, both the peak intracranial pressure (ICP) and average mannitol consumption were significantly reduced compared with those of patients with lower SERPINA3 levels. In terms of the 6-month outcomes measured via the GOS, LCF, DRS, and ERBI, lower levels of SERPINA3 were indicative of poorer prognosis. Furthermore, we found a negative correlation between serum SERPINA3 levels and the VBR. The receiver operating characteristic (ROC) curve and decision curve analysis (DCA) demonstrated the predictive performance of SERPINA3. In conclusion, incorporating the novel biomarker SERPINA3 alongside traditional assessment tools offers neurosurgeons an effective and easily accessible means, which is readily accessible early on, to predict the risk of intracranial pressure elevation and long-term prognosis in TBI patients.

## 1. Introduction

Traumatic brain injury (TBI) is a prevalent condition globally, known for its significant disability, mortality, and complication rates. Over 50 million individuals experience a traumatic brain injury annually, with approximately half of the global population expected to suffer from one or more TBIs during their lifetime [1,2]. TBI is usually caused by mechanical forces applied to the brain, including subdural hematoma, subarachnoid hemorrhage, and brain contusion. Subsequent edema, inflammation, free radical damage, glial scar formation, apoptosis, and necrosis contribute to the development of secondary injuries [3,4]. The initial severity of TBI was usually classified as mild, moderate, or severe based on the clinical awareness level of the Glasgow Coma Scale (GCS) [5]. The GCS score, together with age, computed tomography (CT) findings, hypoxia, and systolic blood pressure are the significant factors affecting the prognosis [6,7,8].

TBI is a multifaceted disorder, and treatment strategies generally relying on guidelines that promote a one-size-fits-all method cannot achieve a satisfactory therapeutic effect. The prediction for the acute phase progression and long-term prognosis of TBI patients is difficult but crucial [9]. Currently, a relatively accurate model for prediction is lacking. The use of simplistic methods hinders the accurate quantification of TBI outcomes and prognosis. The application of multiple new techniques like genomics, blood biomarkers, and pathophysiological monitoring in TBI treatment improves individualized management [10,11]. Diagnostic and prognostic biomarkers for TBI have long been sought over the past decades. Highly specific and sensitive biomarkers play a crucial role in not only unraveling the molecular mechanisms underlying TBI but also guiding the selection of treatment modalities. Classical biomarkers in TBI are released by injured or dying neurons and astrocytes, including neuron-specific enolase (NSE), ubiquitin C-terminal hydrolase-L1 (UCH-L1), glial fibrillary acidic protein (GFAP), S100 calcium binding protein B (S100B), and myelin basic protein (MBP) [12,13,14,15]. These markers provide essential information about the severity of the primary injury, and their elevation is also a reflection of ongoing pathological processes. More reliable biomarkers that correlate with the progression of TBI are needed to advance neuroprotective and therapeutic clinical trials.

Serine protease inhibitor clade A member 3 (SERPINA3), a member of the serpin superfamily, principally works as a protease inhibitor in maintaining homeostasis [16]. Several serine proteases, such as pancreatic chymotrypsin, mast cell chymase, and leukocyte cathepsin G, are inhibited by SERPINA3 [17]. SerpinA3 has been shown to be involved in several pathologies, including cancers, Alzheimer’s disease, neuropathic pain, and stroke [18,19,20,21]. Previous research reported that SERPINA3 is a biomarker involved in diabetic nephropathy renal tubular injury, cerebral small vessel disease, and coronary artery disease [22,23,24]. However, whether or not SERPINA3 could serve as a biomarker for TBI progression and prognosis has not been clearly investigated. In this study, we aimed to delve into the correlation between SEPRINA3, a specific molecule found in increased amounts in TBI patients’ plasma, and the intracranial pressure (ICP) peak, as well as the long-term neurological outcomes after TBI. The commonly used parameters including the GCS and Helsinki score were also evaluated in comparison with SERPINA3 for their prognostic values, and a combined model was established, as the performance and net benefit were compared, aiming to find a more efficient clinical prediction tool for neurosurgeons.

## 2. Materials and Methods

### 2.1. Patients Enrollment

Between February 2020 and December 2021, 37 individuals between the ages of 18 and 65 who had experienced traumatic brain injury (ICD-10 code S06.902) and had surgery at Renji Hospital, Shanghai Jiao Tong University School of Medicine (Shanghai, China), were included in the study. The criteria for inclusion included individuals with serious brain injuries (GCS 3–8) who underwent surgery within 6 h after trauma. Exclusion criteria included various central nervous system conditions, serious systemic diseases, and participants who were not available for follow-up. Strategies for managing TBI involved reducing cerebral edema to lower the intracranial pressure, blood pressure control and preventing hypoxemia, improving cerebral perfusion, avoiding infections, providing sedation and analgesia, and ensuring proper nutrition support. Dehydration treatment was employed to alleviate cerebral edema via utilizing hyperosmolar agents, mainly mannitol, and supplemented with glycerol fructose, hypertonic saline, and furosemide. Mannitol constituted the primary hyperosmolar agent, with individualized combinations chosen based on patient-specific conditions. Postoperative intracranial pressure (ICP) measurements were obtained via ventricular probes, typically stabilizing around 7 days post-surgery. To mitigate the chance of intraventricular infection, monitoring of intracranial pressure was discontinued 10–14 days post-surgery. Additionally, all patients received 1–2 weeks of sedation therapy (dexmedetomidine) as warranted. The study received approval from our hospital’s ethics committee (RA-2020-107), and all participants or their legal representatives provided written informed consent.

### 2.2. Variables

This study gathered data on gender, age, underlying conditions (hypertension, diabetes, or coronary heart disease), preoperative GCS scores, CT scans, SERPINA3 levels, postoperative ICP, daily mannitol dosage during hospital stay, and prognostic scale outcomes 6 months post-surgery.

The medical history of each patient was assessed to ascertain underlying medical conditions. Hypertension, diabetes, and coronary heart disease were diagnosed and managed by specialist physicians using appropriate medications. A diagnosis of hypertension was made when the systolic blood pressure was equal to or greater than 140 mmHg or the diastolic blood pressure was equal to or greater than 90 mmHg for three consecutive mornings without taking antihypertensive medication. Diabetes was confirmed if fasting blood glucose was ≥7.0 mmol/L or two-hour postprandial blood glucose was ≥11.1 mmol/L. Coronary heart disease diagnosis relied on coronary angiography findings indicating more than 50% stenosis.

The Helsinki computed tomography (CT) classification [25] criteria are detailed in Table 1. ICP grading after surgery was determined by the highest value recorded during the ICP monitoring period. Patients were divided into a high-SERPINA3 group and low- SERPINA3 group based on the median SERPINA3 concentration in our study. ICP peaks were categorized as moderately elevated (21–40 mmHg) or severely elevated (>40 mmHg). The Glasgow Outcome Scale (GOS) score [26,27] criteria consist of 5 points for well-recovered individuals returning to normal life despite mild impairment, 4 points for those with mild disability who are disabled but can live independently and work with support, 3 points for severe disability where the individual is awake but reliant on others for daily activities, 2 points for vegetative survival with minimal response like eye opening during cycles of sleep and wakefulness, and 1 point for death. A favorable outcome is defined as a GOS score of 4–5, with patients classified as having poor outcomes if their GOS scores fall between 1 and 3. Further prognostic scales were used to evaluate the rehabilitation of patients 6 months after trauma, including Levels of Cognitive Functioning (LCF) [28], the Disability Rating Scale (DRS) [29], and the Early Rehabilitation Barthel Index (ERBI) [30]. A fixed panel of professionals were present to test the patients according to the scale instructions.

CT scans were taken before and 6 months after surgery to collect data, which involved calculating parameters such as the intra-cranial volume (ICV), Ventricle volume (VV), and ventricle-to-intracranial-volume ratio (VBR). The ventricle-to-intra-cranial-volume ratio (VBR) was adjusted for differences in head size using the following calculation [31,32]:VV/ICV×1000.

### 2.3. ELISA Analysis

The concentration of SERPINA3 in serum was determined with an ELISA kit (KA2128) from Abnova in Taiwan, China. Peripheral venous blood samples were obtained from patients before surgery, followed by centrifugation to isolate the supernatant. The ELISA assay was then performed according to the manufacturer’s instructions. Each blood sample underwent three independent tests, and the average of these results was calculated to determine the final SERPINA3 level.

### 2.4. Statistical Methods

The statistical analysis was performed with the SPSS 26.0 and R (4.2.2) software. Numeric variables were summarized using the mean ± SD for normally distributed data, or the median for non-normally distributed data. Frequencies were used to present categorical variables. The Spearman correlation test was employed for correlation analysis. ROC curves and the AUC were used to evaluate how well specific parameters and composite models performed. Furthermore, the model’s net benefit for patients was assessed using decision curve analysis (DCA) [33]. A *p*-value of less than 0.05 was used to determine statistical significance.

## 3. Results

### 3.1. Baseline Patient Characteristics

Over the course of the study, there were 239 TBI patients in total who were finally hospitalized with clinical records. From the entire patient pool, 121 patients (50.63%) opted for conservative treatment, and only surgical treatment was administered to 41 patients (17.15%) without the placement of intracranial pressure probes. Additionally, data for the 6-month follow-up were missing for 40 patients (16.74%). As shown in Table 2, we finally enrolled a number of 37 patients in this study, who all experienced a standard surgical treatment procedure. In total, 17 out of 37 patients showed severe ICP elevation in the NICU, and there were 17 patients with unfavorable outcomes of GOS (GOS 1 to 3). When comparing the baseline characteristics, we found no significant differences between the two subgroups concerning gender, age, hypertension, diabetes mellitus, or coronary heart disease.

### 3.2. The Correlation between Serum SERPINA3 Level and Severity of TBI

The serum SERPINA3 concentration was compared in patients with different GCS scores, while no significant difference was observed (Figure 1a). However, when comparing serum SERPINA3 concentrations between patients with severe (GCS 6–8 points) and critical (GCS 3–5 points) GCS scores, we observed a significant decrease in serum SERPINA3 levels among patients in the critical TBI group compared with those in the severe TBI group (Figure 1b). Furthermore, we assessed Helsinki CT scores based on imaging findings in patients. Upon comparing serum SERPINA3 levels across different Helsinki CT scores, it became evident that patients with higher scores tended to have decreased serum SERPINA3 concentrations (Figure 1c).

### 3.3. Preoperative Serum SERPINA3 Level Indicated Intracranial Hypertension Risk in Neuro-Intensive Care Unit

Figure 2a exhibits the correlation between serum SERPINA3 concentrations and peak ICP levels among the patient cohort, whereas Figure 2b showcases the relationship between serum SERPINA3 levels and the average mannitol usage throughout hospitalization. Patients were segregated into two groups based on their postoperative peak ICP readings: those with moderately high ICP levels (ranging from 21 to 40 mmHg) and those with severely high ICP levels (exceeding 40 mmHg). Notably, within our study, 18 patients encountered severe elevations in ICP, comprising 48.65% of the total samples. Upon comparing serum SERPINA3 concentrations between patients with moderately high and severely high ICP levels, we discerned a noteworthy decrease in serum SERPINA3 levels among those belonging to the latter group compared with those of patients in the former (Figure 2c). Furthermore, upon stratifying patients based on their SERPINA3 levels and examining the peak ICP levels and average mannitol usage across groups with high and low serum SERPINA3 concentrations, a substantial disparity emerged, with significantly lower peak ICP levels and mannitol consumption observed in the high-SERPINA3 group in contrast to the low-SERPINA3 group (Figure 2d,e). Simultaneously, when considering both SERPINA3 levels and peak ICP and analyzing ICP levels between patients with high and low SERPINA3 concentrations, an insightful observation was made: while 26.3% of patients in the high-SERPINA3 group experienced severe ICP elevations (exceeding 40 mmHg), the majority (73.7%) encountered moderate ICP elevations (ranging from 21 to 40 mmHg). Conversely, in the low-SERPINA3 group, a notable proportion (66.7%) exhibited severe ICP elevation, with a smaller fraction (33.3%) experiencing moderate ICP elevation (Table 3).

### 3.4. Long-Term Structural Changes on CT Images Correlate with Preoperative Serum SERPINA3 Levels

Concerning the VBRs, upon stratifying patients based on their SERPINA3 levels and examining the VBR across groups with high and low serum SERPINA3 concentrations, a notable difference emerged: a significantly lower VBR was observed in the high-SERPINA3 group compared with the low SERPINA3 group at 6 months post-trauma (*p* < 0.05), as illustrated in Figure 3b. However, the difference in VBRs measured based on CT scans obtained preoperatively was not significant (Figure 3a and Figure 4). Furthermore, our analysis revealed a negative correlation between the VBR and SERPINA3 concentration at 6 months post-trauma (*p* < 0.01), as depicted in Figure 3c and Figure 4.

### 3.5. The Significance of Serum SERPINA3 Level in Predicting Intracranial Hypertension Risk and Poor Neurologic Outcomes

Correlation analysis revealed a significant association between SERPINA3 levels and prognostic scales such as DRS, ERBI, and GOS, while the correlation with the LCF score was not significant (Table 4). Furthermore, we explored the predictive value of SERPINA3 levels concerning intracranial hypertension risk and long-term neurological outcomes. Employing receiver operating characteristic (ROC) curve analysis, we assessed SERPINA3 concentrations in conjunction with GCS scores and Helsinki scores. As depicted in Figure 5a, SERPINA3 concentrations exhibited discriminative potential in differentiating levels of ICP elevation. Notably, the Helsinki score demonstrated the highest performance, achieving an AUC of 0.8044. Subsequently, the SERPINA3 concentration outperformed the GCS score, yielding an AUC of 0.8618, while the AUC for GCS scores was 0.7794. The combined model demonstrated the best predictive performance, with an AUC value of 0.9125. Concerning the prediction of poor neurological outcomes, the SERPINA3 concentration displayed an AUC of 0.7202, with the Helsinki score once again surpassing other parameters with an AUC of 0.7351 (Figure 5c). Additionally, the DCA curve visually demonstrated that the combined use of the GCS score, SERPINA3, and the Helsinki score conferred a substantial net benefit in predicting both ICP elevation risks and long-term neurological outcomes compared with alternative models across the relevant threshold range in our study cohort (Figure 5b,d).

## 4. Discussion

Traumatic brain injury (TBI) stands as one of the most prevalent traumatic conditions worldwide, often necessitating surgical intervention as the primary treatment. The acute phase of TBI unfolds rapidly, characterized by intracranial pressure (ICP) elevation that can lead to brain herniation—a significant contributor to mortality [34]. Additionally, predicting long-term neurological outcomes in TBI patients pose a challenge due to the multifactorial nature of the condition [35,36,37,38]. In our study, we sought to explore the role of serum SERPINA3 concentrations in predicting the progression of ICP and poor neurological outcomes in TBI patients. Our primary finding suggests that SERPINA3 concentrations, gleaned from preoperative blood samples, could effectively stratify TBI patients based on their likelihood of experiencing high ICP elevation or poor neurological outcomes, with respective AUCs of 0.8618 and 0.7202. Furthermore, our results revealed a close association between serum SERPINA3 levels and the 6-month VBR post-trauma. Notably, combining the GCS score, SERPINA3 concentration, and Helsinki score significantly enhanced predictive performance in TBI patients, yielding increased AUCs both in predicting ICP issues and long-term outcomes, alongside heightened net benefits in the DCA curves. These findings underscore the predictive potential of SERPINA3 in assessing ICP progression and unfavorable outcomes in the context of TBI.

In recent years, clinical investigations into SERPINA3 have garnered substantial attention across various domains. Research indicates a pivotal role for SERPINA3 in neurodegenerative disorders, with findings suggesting its association with the pathogenesis and neural degeneration in Alzheimer’s disease, potentially offering novel therapeutic targets, with recent research showing that SERPINA3 was deposited in amyloid plaques in Alzheimer’s disease (AD) [39,40], elevated in AD patient plasma [41] and proposed to regulate the disease’s pathophysiology [41,42]. Furthermore, insights reveal a connection between SERPINA3 and cardiovascular diseases, such as its correlation with the progression of coronary artery disease [24], and its significant correlation with the clinical outcomes in myocardial infarction as well as non-ischemic cardiomyopathies [43], suggesting its utility as a biomarker. Additionally, SERPINA3’s involvement in inflammation and immune regulation has emerged as a significant area of interest, exemplified by its elevated expression in stroke patients, correlating with cerebral small vessel disease and validating its relationship with inflammation and endothelial dysfunction [23]. These investigations provide crucial insights into the biological functions of SERPINA3 and its mechanistic roles in disease progression, laying the groundwork for the development of future therapeutic modalities and diagnostic tools.

Given that intracranial hypertension significantly influences prognosis, the prevention and management of elevated ICP levels are pivotal components of targeted treatment strategies post-surgery [44]. Dehydration stands as a common intervention for high ICP levels; however, postoperative changes in intracranial pressure occur rapidly, posing challenges in determining the optimal timing for dehydration therapy [34]. Consequently, intracranial pressure may escalate to uncontrollable levels, severely impacting patient survival and recovery [44]. However, only 36.23% of TBI patients underwent ICP monitoring in China [45], and in resource-limited regions, including medical resource-limited small towns, disaster sites, and war zones, the scarcity of equipment and intensive care unit (ICU) resources may result in even lower ratios. In such contexts, SERPINA3, can aid surgeons in resource allocation and treatment planning. This may involve prioritizing the use of ICP monitoring for patients at high risk of elevated ICP, implementing more aggressive early decompressive craniectomy for high-risk patients, and administering higher doses of dehydration drugs in advance. Further, in a normal ICU, implementing SERPINA3 in clinical practice could also lead to a more proactive approach to managing patients. As of now, although numerous studies suggest that SERPINA3 serves as a protective factor in acute central nervous system injuries such as TBI and stroke, its specific mechanism of action remains to be fully elucidated. Former research has demonstrated that SERPINA3 exerts its protective role in BBB integrity by inhibiting the infiltration of inflammatory cells and the activity of proteases such as GZMB [46,47]. Disruption of the blood–brain barrier (BBB) stands out as a key factor in the onset and progression of postoperative cerebral edema [48], and dramatically, our study revealed a close association between SERPINA3 expression and ICP levels, indicating that patients with elevated SERPINA3 levels exhibited reduced severity of postoperative intracranial hypertension and displayed enhanced responsiveness to dehydration therapy. Thus, our findings may also be attributed, in part, to the protective effect of SERPINA3 on the integrity of the blood–brain barrier.

In our research, SERPINA3 concentrations were associated with better prognosis when calculating the VBR, GOS score, ERBI, and DRS score. Emerging research indicates that after the resolution of edema and/or hemorrhage post-TBI [49,50], a change in total brain volume typically occurs approximately 3 weeks post-injury, accompanied by an expansion in ventricular volume [49,50]. This enlargement of the ventricles beyond pre-injury levels is attributed to the vacated space resulting from the atrophy of damaged brain tissue [49,50,51]. However, excessive increases in ventricular volume often indicate the development of acquired hydrocephalus. The degree of ventricular enlargement closely correlates with the severity of brain tissue atrophy, with larger ventricles indicating more pronounced tissue loss. An intriguing observation is that recent studies have demonstrated that the upregulation of SERPINA3 confers neuroprotection, attenuating neuronal injury and preventing loss in both ischemic and TBI [19,47,52]. On the other hand, hydrocephalus is often associated with various CNS disorders [53,54,55]. In line with the hypothesis suggesting that acquired hydrocephalus may have an immune-mediated component, the occurrence of hydrocephalus was notably reduced in a cohort of patients who received dexamethasone following subarachnoid hemorrhage [56]. Similarly, SERPINA3 has been demonstrated to exert anti-inflammatory effects in TBI [47], which might be an additional explanation for the protective effect of SERPINA3. The findings above corroborate the assertion that our conclusions are not merely coincidental, indicating that SERPINA3 might attenuate neuronal loss and inflammatory response in TBI, which further contributed to improving the long-term neurological outcomes.

Predicting long-term outcomes following traumatic brain injury (TBI) presents a significant challenge. Given the rapid onset of TBI, prognostic indicators must be readily obtainable and interpretable. As a result, neurosurgeons commonly rely on peripheral blood biomarkers and imaging scores. To our knowledge, this is the first study to investigate the clinical relevance of SERPINA3 in TBI. Recent reports have highlighted the utility of the novel Helsinki CT score in enhancing the accuracy of outcome prediction. The Helsinki CT score presents a feasible alternative to established systems such as the Rotterdam and Marshall CT scoring systems [25]. In our study, we developed a predictive model that integrates the GCS score, Helsinki score, and SERPINA3 levels to assess the risk of elevated ICP and long-term prognosis in TBI patients. Intriguingly, when SERPINA3 and the Helsinki score were evaluated as independent predictors, both exhibited favorable predictive performance. However, the comprehensive model that incorporates all three factors demonstrated superior predictive ability and net benefits for patients.

There are a few limitations to this study. Firstly, it is important to acknowledge that our study was conducted at a single center, limiting the representation of other practice settings. Additionally, the sample size was small, and no long-term follow-up was conducted. Future studies should aim to include a larger sample size and perform multiple comparisons to yield more comprehensive and generalizable conclusions. Secondly, SERPINA3 expression is not exclusively specific to TBI cases [16,18,22,23,43], and more complex situations, such as polytrauma cases, are common in many moderate to severe TBI patients, which could complicate the large-scale use of SERPINA3 as a biomarker. Future research should investigate the effects of SERPINA3 in a polytrauma cohort. Thirdly, to enhance the translational potential of SERPINA3 as a therapeutic target for TBI, it is essential to study comorbidities, such as aging. In future research, we will investigate and discuss patients stratified by different age groups, aligning with current recommendations in the field.

## Figures and Tables

**Figure 1 diagnostics-14-01245-f001:**
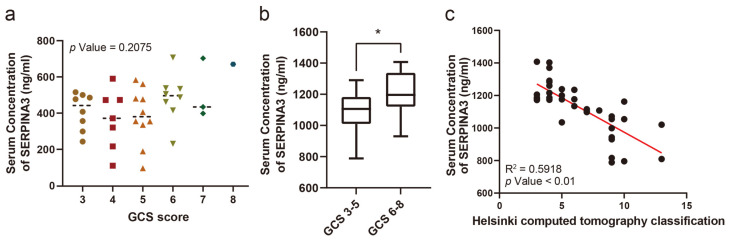
The correlation between the serum SERPINA3 level and the severity of traumatic brain injury. (**a**) A comparison of serum SERPINA3 levels among patients with different Glasgow Coma Scale scores. (**b**) The different distributions of concentrations of serum SERPINA3 in the critical traumatic brain injury group (GCS 3–5) and severe traumatic brain injury group (GCS 6–8). (**c**) The correlation between serum SERPINA3 levels and Helsinki computed tomography classification scores. * *p* Value < 0.05.

**Figure 2 diagnostics-14-01245-f002:**
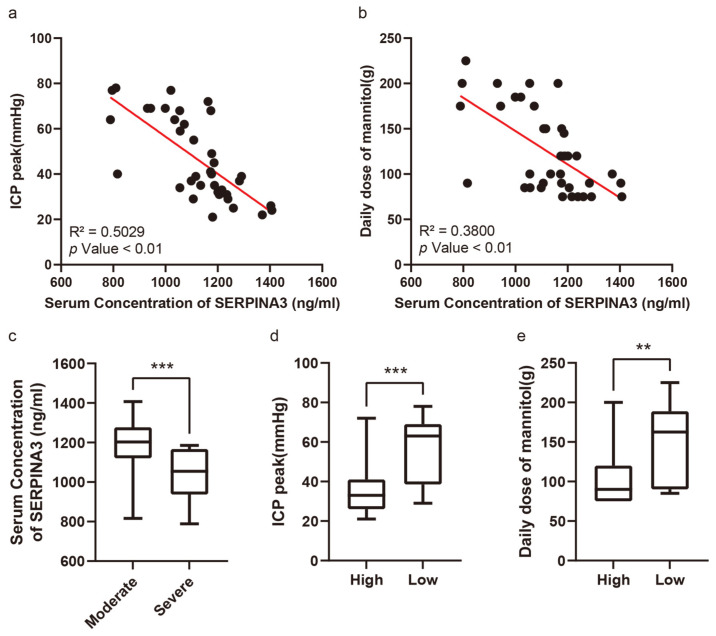
The serum SERPINA3 level serving as an indicator of postoperative intracranial pressure and sensitivity to dehydration therapy. (**a**) The correlation between serum SERPINA3 levels and intracranial pressure peaks. (**b**) The correlation between serum SERPINA3 levels and the use of a daily dose of mannitol. (**c**) The different distributions of concentrations of serum SERPINA3 in the moderate ICP elevation group and severe ICP elevation group. (**d**) The different distributions of ICP peak levels in the high-SERPINA3 group and low-SERPINA3 group. (**e**) The different distributions of the use of a daily dose of mannitol in the high-SERPINA3 group and low-SERPINA3 group. ** *p* Value < 0.01, *** *p* Value < 0.001.

**Figure 3 diagnostics-14-01245-f003:**
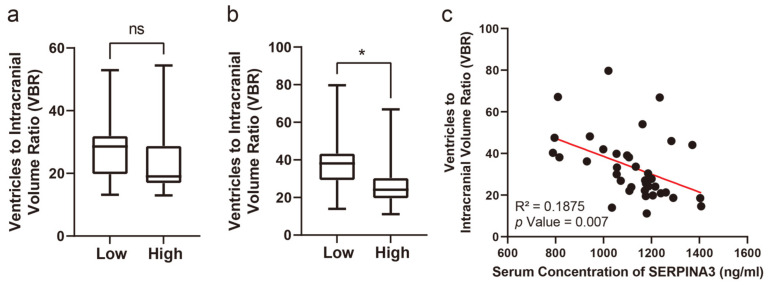
Serum SERPINA3 levels indicating the VBR during the long-term recovery of patients with TBI. (**a**) The different distributions of preoperative VBRs in the high-SERPINA3 group and low-SERPINA3 group. (**b**) The different distributions of the 6-month postoperative VBRs in high-SERPINA3 group and low-SERPINA3 group. (**c**) The correlation between serum SERPINA3 levels and the 6-month postoperative VBR. * *p* Value < 0.05, ns *p* Value > 0.05.

**Figure 4 diagnostics-14-01245-f004:**
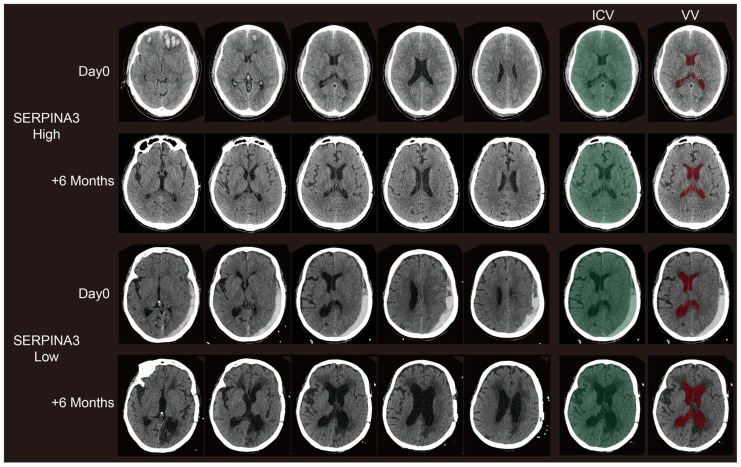
Representative paired CT scans of patients in high-SERPINA3 group (the first and second lines) and-low SERPINA3 group (the third and fourth lines) at day 0 (the first and third lines) and 6 months postoperatively (the second and fourth lines). The ICV column stands for the selection of the intracranial volume (ICV). The VV column stands for the selection of the ventricle column (VV). The selection of both the ICV and VV were controlled via visual inspection.

**Figure 5 diagnostics-14-01245-f005:**
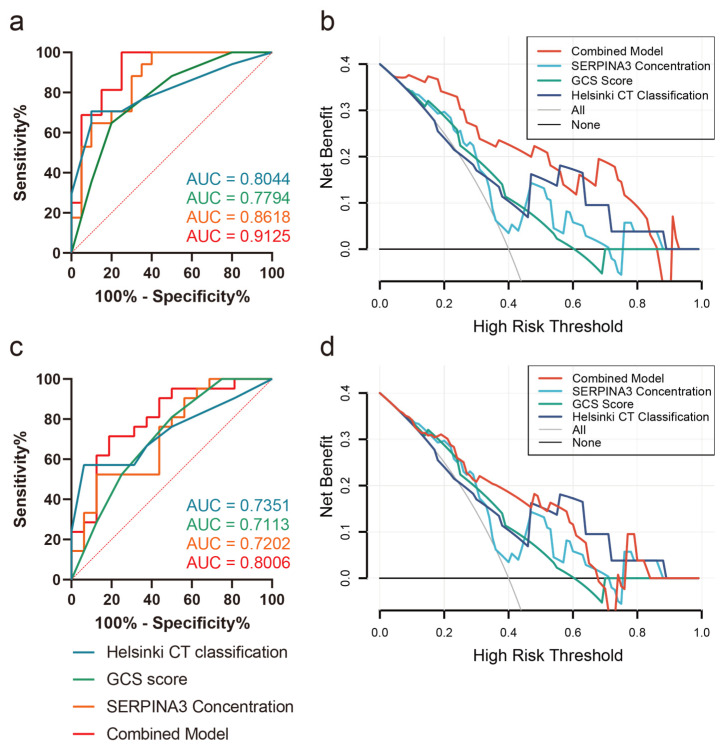
Receiver operating characteristic curves (ROCs) and decision curve analysis (DCA) curves (**a**) Prediction of ICP peak levels; (**b**) DCA curve of ICP peak levels; (**c**) prediction of GOS outcomes 6 months postoperatively; (**d**) DCA curve of GOS outcomes 6 months postoperatively.

**Table 1 diagnostics-14-01245-t001:** Helsinki computed tomography (CT) classification.

Component	Description	Score
Mass lesion type	Subdural Hematoma	2
	Contusion	2
	Intracerebral Hematoma (Parenchymal)	2
	Epidural Hematoma	−3
Mass lesion size	Hematoma Volume > 25 cm^3^	2
Intraventricular hemorrhage	Absent	0
	Present	3
Suprasellar cistern	Normal	0
	Compressed	1
	Obliterated	5
Sum score	Range: −3 to 14	

**Table 2 diagnostics-14-01245-t002:** The baseline characteristics of the patients.

Characteristics	Overall	ICP Level			GOS Score		
		Moderate	Severe	*p* Value	Favorable	Unfavorable	*p* Value
*n*	37	20	17		16	21	
Gender (male, %)	16 (43.2)	10 (50.0)	6 (35.3)	0.571	8 (50.0)	8 (38.1)	0.697
Age (mean (SD))	44.84 (12.33)	44.10 (11.53)	45.71 (13.51)	0.699	45.69 (10.38)	44.19 (13.84)	0.72
Hypertension (%)	11 (29.7)	5 (25.0)	6 (35.3)	0.748	3 (18.8)	8 (38.1)	0.362
Diabetes mellitus (%)	12 (32.4)	8 (40.0)	4 (23.5)	0.475	6 (37.5)	6 (28.6)	0.826
Coronary heart disease (%)	9 (24.3)	5 (25.0)	4 (23.5)	1	5 (31.2)	4 (19.0)	0.638

ICP: intracranial pressure; GOS: Glasgow Outcome Scale.

**Table 3 diagnostics-14-01245-t003:** Comparison of ICP peak levels in different groups.

SERPINA3 Group	ICP Elevation Level (mmHg)		Total	*p* Value
	21–40	>40		
Low	6	12	0.0138	
High	14	5		

ICP: intracranial pressure.

**Table 4 diagnostics-14-01245-t004:** Correlation between serum SERPINA3 levels and different prognostic scales.

	Serum SERPINA3 Level			
	LCF	DRS	ERBI	GOS
Spearman r	0.2506	−0.3863	0.3991	0.4701
*p* Value	0.1347	0.0182	0.0144	0.0033

LCF: Levels of Cognitive Functioning; DRS: Disability Rating Scale; ERBI: Early Rehabilitation Barthel Index; GOS: Glasgow Outcome Scale.

## Data Availability

All study data are available from the corresponding author by e-mail.

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
