# Peer review of "Prediction Value of Initial Serum Levels of SERPINA3 in Intracranial Pressure and Long-Term Neurological Outcomes in Traumatic Brain Injury"

_diagnostics, 2024, doi:10.3390/diagnostics14121245_

Round 1

Reviewer 1 Report

Comments and Suggestions for Authors

The current manuscript by Tan et al. sought to explore the predictive value of serum SERPINA3 levels in identifying the risk of cerebral edema and prognostic value in the context of TBI. The study consisted of 37 patients with TBI and outcome scores across various applicable clinical measures at 6 months post injury. Lower levels of SERPINA3 were found to be indicative of a poorer prognosis. The manuscript is timely, has an amount of novelty associated with it and may represent a useful clinical tool for the early assessment of outcomes in moderate to severe TBI. There are some issues however that should be addressed prior to publication of the manuscript however. These are outlined below:

1.    The beginning of the second sentence of the introduction is missing a space prior to the sentence.

2.    Within the same sentence there is an errant space within the middle of the sentence.

3.    The first sentence of the second paragraph within the Introduction reads, “…thus making prediction for the acute phase progression and long-term prognosis of TBI patients is difficult but crucial[9].” This reads sort of odd. Reviewer recommends an editor go through the manuscript and fix typos and/or English language irregularities.

4.    Within Figure 2, the patients are delineated by “high” and “low” SERPINA3 levels. Is there a numerical value that was chosen with regards as to what constitutes High or Low values in SERPINA3 levels?  The levels for ICP are clearly displayed within the results section however the levels for SERPINA3 itself appear to be missing from the manuscript.

5.    Although the data is certainly impressive and certainly warrants further investigation, SERPINA3 levels do not appear to be specific for TBI alone (by the authors own admission with references included), however this is not noted a significant limitation of the current study. It is possible that SERPINA3 levels correlate to outcomes, however it is not solely specific to TBI cases and make it use clinically problematic when it comes to various forms of polytrauma. Although this isn’t a fatal flow by any means, it should certainly be noted as a limitation within the discussion of the current manuscript. Many clinical cases of moderate to severe TBI are complicated polytrauma cases and could make the large-scale implementation of SERPINA3 as a biomarker problematic. On the other hand, it could also increase the validity of it as a useful clinical marker. This should be noted.

Comments on the Quality of English Language

English language needs editing in certain places of the manuscript. 

Reviewer 2 Report

Comments and Suggestions for Authors

SERPINA3 is a potential biomarker for assessing this inflammation. This study aimed to explore the predictive value of post-operative serum SERPINA3 levels in identifying the risk of cerebral edema and its prognostic implications in traumatic brain injury.  The study included patients with Glasgow Coma Scale (GCS) scores ranging from 3 to 8.  Among patients with elevated SERPINA3 levels, both the peak intracranial pressure (ICP) and average mannitol consumption were significantly reduced compared to those with lower SERPINA3 levels.   Lower levels of SERPINA3 were indicative of a poorer prognosis. SERPINA3 appears to serve as a protective role in a variety of neurologic conditions including head injury and stroke presumably by inhibiting the infiltration of inflammatory cells.  It is important to note that no long-term follow-up was conducted in this study. This is an interesting study although the clinical relevance is somewhat limited.  Every patient should get the maximal treatment, and it is not particularly helpful to know that the outcome is likely to be poor in patients with a low GCS regardless of the SERPINA3 level. Some discussion of the clinical management issues determined by the SERPINA3 levels should be incorporated in this paper.

Comments on the Quality of English Language

Minor editing indicated.

Reviewer 3 Report

Comments and Suggestions for Authors

This study is interesting. However, I have some suggestions.

Research on SERPINA3 has been conducted in relation to inflammation, and there are studies related to neuroinflammation as well. However, it still prompts doubts about whether it can be considered a neurologically specific marker in TBI.

The patient numbers in the text and the table are confusing. The manuscript indicates “Eighteen out of 37 patients showed severe ICP elevation in NICU and there were 17 patients with unfavorable outcomes of GOS (GOS 1 to 3). Table 2 indicates that out of 37 patients, 20 are classified as moderate and 17 as severe. Please verify it.

In the methods, the age range for participants is specified as 18 to 65 years old who had experienced TBI. While there are no significant differences in mean age among the groups, it still leaves me curious about whether there was any practical impact, despite age being closely related to GCS and neurological outcomes.

In Figure 1, there is a concern that the increased concentration of SERPINA3 in the GCS 6-8 group may be disproportionately influenced by one particularly high sample concentration at each score point.

Round 2

Reviewer 3 Report

Comments and Suggestions for Authors

Please ensure the final revisions are completed successfully. Thank you.